# Bichromatic Splicing Detector Allows Quantification of *THRA1* and *THRA2* Splicing Isoforms in Single Cells by Fluorescent Live-Cell Imaging

**DOI:** 10.3390/ijms252413512

**Published:** 2024-12-17

**Authors:** Eugenio Graceffo, Elisa Pedersen, Marta Rosário, Heiko Krude, Markus Schuelke

**Affiliations:** 1Department of Neuropediatrics, Charité-Universitätsmedizin Berlin, Corporate Member of Freie Universität Berlin, Humboldt-Universität Berlin, and Berlin Institute of Health, 13353 Berlin, Germany; eugenio.graceffo@charite.de; 2Einstein Center for Neurosciences Berlin, Charité-Universitätsmedizin Berlin, Corporate Member of Freie Universität Berlin, Humboldt-Universität Berlin, and Berlin Institute of Health, 10117 Berlin, Germany; elisa.pedersen@charite.de; 3Institute of Cell and Neurobiology, Charité-Universitätsmedizin Berlin, Corporate Member of Freie Universität Berlin, Humboldt-Universität Berlin, and Berlin Institute of Health, 10117 Berlin, Germany; marta.rosario@charite.de; 4Department of Pediatric Endocrinology, Charité-Universitätsmedizin Berlin, Corporate Member of Freie Universität Berlin, Humboldt-Universität Berlin, and Berlin Institute of Health, 13353 Berlin, Germany; heiko.krude@charite.de; 5Neurocure Clinical Research Center, Charité-Universitätsmedizin Berlin, Corporate Member of Freie Universität Berlin, Humboldt-Universität Berlin, and Berlin Institute of Health, 10117 Berlin, Germany

**Keywords:** thyroid hormone receptor alpha, alternative splicing, splicing reporter, antisense oligonucleotide, life-cell imaging, fluorescent labeling

## Abstract

Thyroid hormone receptor alpha (THR*α*) is a nuclear hormone receptor that binds triiodothyronine (T3) and acts as an important transcription factor in development, metabolism, and reproduction. The coding gene, *THRA*, has two major splicing isoforms in mammals, *THRA1* and *THRA2*, which encode THR*α*1 and THR*α*1, respectively. The better characterized isoform, THR*α*1, is a transcriptional stimulator of genes involved in cell metabolism and growth. The less well-characterized isoform, THR*α*2, lacks the ligand-binding domain (LBD) and may act as an inhibitor of THRα1 activity. Thus, the ratio of THR*α*1 to THR*α*2 isoforms is critical for transcriptional regulation in various tissues and during development and may be abnormal in a number of thyroid hormone resistance syndromes. However, the complete characterization of the THRα isoform expression pattern in healthy human tissues, and especially the study of changes in the ratio of THRα1 to THRα2 in cultured patient cells, has been hampered by the lack of suitable tools to detect the isoform-specific expression patterns. Therefore, we developed a plasmid *pCMV-THRA-RFP-EGFP* splicing detector that allows the visualization and quantification of the differential expression of *THRA1* and *THRA2* splicing isoforms in living single cells during time-lapse and perturbation experiments. This tool enables experiments to further characterize the role of THRα2 and to perform high-throughput drug screening. Molecules that modify *THRA* splicing may be developed into drugs for the treatment of thyroid hormone resistance syndromes.

## 1. Introduction

The thyroid hormone receptor alpha (THRα) serves as a nuclear hormone receptor specifically responsive to triiodothyronine (T3) and plays a central role in various physiological processes throughout the animal kingdom. As a transcription factor, THRα regulates a large number of target genes involved in development, metabolism, and growth [1]. In mammals, two major isoforms are produced by alternative splicing: THRα1 (encoded by *THRA1*), which carries a T3-sensitive ligand-binding domain (LBD), and THRα2 (encoded by *THRA2*), which lacks the T3 binding site, rendering its activity independent of the presence of T3 (Figure 1a) [2]. In the absence of T3, THRα1 binds to thyroid hormone-responsive elements (TREs) located in the promoter regions of target genes, resulting in repression of gene expression [3]. Upon T3 binding to THRα1, structural changes in the receptor facilitate coactivator binding, ultimately leading to transcription of target genes. In contrast, THRα2, which lacks the ability to bind T3, has been shown to function as a weak dominant-negative inhibitor of THRα1, possibly through protein–protein interactions [4]. As a result, the balance between THRα1 and THRα2 concentrations dictates the cellular response to T3; a higher ratio of THRα1 to THRα2 may be associated with a more pronounced T3 response and target gene activation, whereas an increased presence of THRα2 may correlate with decreased target gene expression (Figure 1b). Bioinformatic analyses have provided preliminary evidence for a tissue- and time-specific expression of these two isoforms, with a predominance of *THRA1* mRNA in skeletal muscle and adipose tissue, and a strong predominance of *THRA2* mRNA in the developing healthy human brain and specifically in cell populations that are at a later stage of differentiation [5].

The importance of aberrations in splicing and thus in the response to T3 becomes apparent when considering mutations within the coding sequences of the human *THRA* gene [6,7,8]. The majority of these mutations disrupt the binding of T3 to THRα1, thereby increasing the pool of receptors that are unresponsive to T3. This results in T3 resistance, characterized by a severe phenotype of congenital non-goitrous hypothyroidism and neuromotor developmental delay (OMIM #614450). Clinical manifestations include developmental delay, intellectual disability, failure to thrive, bradycardia, delayed ossification, and an overall reduction in the metabolic rate [7,8,9]. It can be speculated that difficult-to-detect mutations in the non-coding splicing regulatory regions of *THRA*, resulting in an aberrant ratio of THRα1 to THRα2 isoforms, may cause additional as-yet-undiagnosed inherited diseases associated with the *THRA* gene.

Importantly, the study of such alterations in patient-derived in vitro cultures and, more generally, the full characterization of *THRA* splicing isoform expression patterns in healthy human tissues have been hampered by the lack of a suitable molecular tool to detect isoform-specific expression of THRα1 and THRα2. To address this technological gap, we have developed a plasmid splicing detector (*pCMV-THRA-RFP-EGFP*) that allows in vitro visualization of *THRA1* and *THRA2* splicing events in living cells. Specifically, the splicing event that produces the THRα1 isoform results in the expression of red fluorescent protein (RFP), while the splicing event forming the THRα2 isoform leads to the expression of enhanced green fluorescent protein (EGFP). A schematic representation of the detection mechanism is shown in Figure 1c. The *pCMV-THRA-RFP-EGFP* splicing detector can be used to compare expression levels of *THRA* splicing isoforms between different in vitro conditions (e.g., healthy versus disease, control versus treated), to co-detect and co-quantify the ratio of THRα1 to THRα2 at single cell resolution in time-lapse experiments, and for further downstream processes such as FACS sorting.

## 2. Results

### 2.1. pCMV-THRA-RFP-EGFP Splicing Detector Design and Cloning

To study the relative expression levels of *THRA1* and *THRA2* splicing isoforms in vitro, we generated a DNA plasmid expressing an RFP fusion protein as a proxy for *THRA1* splicing and an EGFP fusion protein as a proxy for *THRA2* splicing.

First, we selected the genomic region (including introns) of *THRA* spanning from exon 7 to the 3′ UTR site of exon 10 (GRCh38 chr17: 40,086,707–40,093,867; Figure 2a). We chose this region of 7.3 kbp based on previous results from the Monroe laboratory [10,11,12] and specifically to preserve the splicing regulatory regions around exon 9. We generated PCR primers to amplify the region of interest, thereby adding a *Bgl*II restriction site and a Kozak sequence upstream of exon 7 and a *Not*I restriction site downstream of the 3′ UTR region. To reduce PCR artifacts, we PCR amplified the resulting region of interest using a proofreading expand-PCR system. Figure 2b shows the band corresponding to the 7.3 kbp PCR product on an agarose gel. We then cloned this PCR product into the pGEM-T Easy Vector for amplification. Second, to express our region of interest in mammalian cells, we used restriction enzyme cloning to insert our region of interest into an expression vector containing the human cytomegalovirus (CMV) promoter. Figure 2c shows the two bands corresponding to the 4 kbp linearized expression vector and the 7.3 kbp gene region of interest on an agarose gel.

Finally, we used a modified version of the Megaprimer PCR protocol [13] to replace the stop codon of *THRA* exon 9b with an RFP insert and the stop codon of *THRA* exon 10 with an EGFP insert. Both color tags retained their natural termination codons. After intracellular splicing and translation, we thereby obtained an N-terminally truncated THRα1 tagged with RFP and a truncated THRα2 tagged with EGFP. Figure 2d shows a graphical representation of the final plasmid, hereafter referred to as the *pCMV-THRA-RFP-EGFP* splicing detector.

### 2.2. Validation of the pCMV-THRA-RFP-EGFP Splicing Detector In Vitro

To test whether the modified *THRA* region in our *pCMV-THRA-RFP-EGFP* splicing detector is correctly spliced and translated in vitro in the same way as endogenous *THRA*, we transfected fibroblast-like COS1 cells and N2A neuroblastoma cells with our *pCMV-THRA-RFP-EGFP* splicing detector. Live-cell imaging after 48 h detected both RFP as a proxy for THRα1 and EGFP as proxy for THRα2 in both cell types (Figure 3a). We obtained transfection efficiencies of 64% and 47%, respectively (Figure 3b, n = 15 images). An important feature of our *pCMV-THRA-RFP-EGFP* splicing detector is its ability to quantify the ratio of alternative splicing in individual cells using quantitative fluorescence microscopy. Fluorescence intensity for individual cells was quantified using the QuPath software v0.5.1 [14] and the results are presented as percent fluorescence intensity in the red and green fluorescence channels. The *pCMV-THRA-RFP-EGFP* splicing detector revealed a slightly higher THRα1 than THRα2 expression in COS1 cells (Figure 3c, 52% RFP and 48% EGFP, n = 15) and a strong predominance of THRα2 over THRα1 expression in N2A cells (77% EGFP and 23% RFP, n = 12). Furthermore, out of all transfected COS1 cells, 30% expressed both EGFP and RFP (Figure 3d, COS1 white column, 30.0 ± 0.1%, n = 15), 53% expressed only RFP (COS1 magenta column, 53.0 ± 0.1%, n = 15), and a mere 17% expressed only EGFP (COS1 green column, 17.0 ± 0.1%, n = 15). On the other hand, out of all transfected N2A neuroblastoma cells, 48% expressed both EGFP and RFP (Figure 3d, N2A white column, 48.0 ± 0.2%, n = 12), 45% expressed only EGFP (N2A green column, 45.0 ± 0.2%, n = 12), and 8% expressed only RFP (N2A magenta column 8.0 ± 0.1%, n = 12).

After live-cell imaging, we extracted and quantified the RNA levels of RFP (*THRA1* proxy) and EGFP (*THRA2* proxy) by RT-qPCR and compared them with the RNA levels of exon 9b (*THRA1* specific) and exon 10 (*THRA2* specific) in naïve, untransfected COS1 and N2A neuroblastoma cells, respectively (Figure 3e,f). As expected, the normalized mRNA expression levels of the CMV-driven RFP and EGFP transcripts in both transfected cell types were more than two orders of magnitude higher than the endogenously driven exon 9b and exon 10 mRNA transcripts in untransfected naïve samples (Figure 3e,f, green and magenta columns versus light and dark blue columns). More importantly, the RFP-to-EGFP ratios in *pCMV-THRA-RFP-EGFP* splicing detector-transfected samples appeared to reflect the endogenous ratio of *THRA1* to *THRA2* in naïve samples, with COS1 cells showing a slight predominance of *THRA1*/RFP (Figure 3e) and N2A neuroblastoma cells showing a predominance of *THRA2*/EGFP (Figure 3f), as previously measured by fluorescence signals (Figure 3c).

### 2.3. The pCMV-THRA-RFP-EGFP Splicing Detector Reports ASO-Mediated Splicing Perturbations

To test whether the *pCMV-THRA-RFP-EGFP* splicing detector can be used in vitro as a tool to detect differences in the *THRA* splicing pattern in perturbed/diseased systems, we sequentially transfected COS1 cells first with antisense oligonucleotides (ASOs) that were designed to block the exon 9/10 splicing by binding to the exon 10 splice acceptor site and then with the *pCMV-THRA-RFP-EGFP* splicing detector. This was to prevent the expression of the THRα2 isoform. A schematic representation of the mechanism of action of the ASO blockers is shown in Figure 4b. Figure 4a shows example images of the three conditions: COS1 cells transfected with the *pCMV-THRA-RFP-EGFP* splicing detector (SD) alone (control), COS1 cells treated with a scrambled control ASO and transfected with the *pCMV-THRA-RFP-EGFP* splicing detector (scramble + SD), and COS1 cells treated with *THRA2*-specific ASO and transfected with the *pCMV-THRA-RFP-EGFP* splicing detector (ASO + SD). As expected, we observed a significant knockdown of the EGFP signal (THRα2 proxy) in the ASO-treated samples compared to the *pCMV-THRA-RFP-EGFP* splicing detector-only control. This resulted in a shift in the RFP-to-EGFP ratio from 52 to 48% in controls to 100% RFP and 0% EGFP in treated COS1 cells (Figure 4c, Kruskal–Wallis test, Dunn’s correction for multiple comparisons, n = 15 POVs per condition, *p* < 0.01). As a control, samples treated with scrambled control ASO showed no difference in EGFP and RFP expression levels compared to the *pCMV-THRA-RFP-EGFP* splicing detector-only control. Additionally, we pretreated transfected cells with a CY5-tagged scrambled control to check for proper ASO uptake (Figure 4d). Since 98.4% of the treated samples were CY5^+^, it can be safely assumed that all EGFP^+^/RFP^+^ cells treated with ASO do indeed contain the *THRA2*-specific ASO (Figure 4e, n = 5 POVs) and thus that the observed knockdown of EGFP signal in ASO-treated cells was due to the *THRA2*-specific ASO.

Finally, to ensure that the differences in the measured green and red fluorescence levels indeed reflect the natural ratio of the cell’s endogeneously expressed *THRA1* and *THRA2* transcripts, we extracted RNA from the same samples and performed RT-qPCR on *THRA* exon 7 (proxy for the *pCMV-THRA-RFP-EGFP* splicing detector expression levels) and *THRA* exon 3 (proxy for endogenous *THRA* RNA expression levels). The results showed that *pCMV-THRA-RFP-EGFP* splicing detector expression was comparable in all the *pCMV-THRA-RFP-EGFP* splicing detector-transfected samples regardless of ASO treatment (Figure 4f; ordinary one-way ANOVA and Fisher’s LSD tests; n = 3; ns, not significant; *p* > 0.05) and that the endogenous RNA level of *THRA* was not altered by either *pCMV-THRA-RFP-EGFP* splicing detector introduction and/or ASO treatments (Figure 4g; ordinary one-way ANOVA and Fisher’s LSD tests; n = 3; ns, not significant; *p* > 0.05) compared to naïve samples.

Taken together, these results validate the applicability of our *pCMV-THRA-RFP-EGFP* splicing detector in living cell cultures to co-detect and co-quantify *THRA1* and *THRA2* splicing events in vitro across different treatments/conditions.

### 2.4. pCMV-THRA-RFP-EGFP Splicing Detector Allows Live Quantification of THRα Isoforms In Vitro in Time-Lapse Experiments at Single Cell Resolution

To test the applicability of the *pCMV-THRA-RFP-EGFP* splicing detector in in vitro time-lapse experiments to study the isoform expression pattern of *THRA* over time, we transfected N2A neuroblastoma cells with our *pCMV-THRA-RFP-EGFP* splicing detector and acquired an image every hour for 16 h starting 24 h after the initial transfection (Figure 5a). Similarly, we acquired one image per hour for 48 h from COS1 cells transfected with the *pCMV-THRA-RFP-EGFP* splicing detector starting 24 h after the initial transfection (Figure 6a). In both experimental settings, we were able to manually track individual cells over the course of the experiment and plot the THRα1 and THRα2 expression levels over time (Figure 5b and Figure 6b, n = 10 individual cells per cell type). This allowed us to highlight some cellular variability, where some cells express more THRα1 than THRα2 and vice versa, or some cells switch between THRα1 and THRα2. Additionally, we show an example where it was possible to manually annotate a cell division. Here, the *pCMV-THRA-RFP-EGFP* splicing detector allowed us to associate the specific biological event with THRα1 and THRα2 expression levels (Figure 7, n = 1, zoomed details of Figure 6).

Taken together, these results demonstrate that our *pCMV-THRA-RFP-EGFP* splicing detector can be used in living cell cultures to co-detect and co-quantify THRα isoforms 1 and 2 in vitro in a live time-lapse experiment with single cell resolution.

## 3. Discussion

The importance of *THRA* splicing isoforms and their proper temporal and tissue-specific regulation has recently come into focus [5,8]. However, despite being first described more than 30 years ago, the role of THRα2 remains rather mysterious. This knowledge gap is largely due to the lack of molecular tools to unambiguously quantify THRα1 and THRα2 expression. In fact, the high homology of THRα1 and THRα2 isoforms seems to prevent the production of isoform-specific antibodies. As a result, the only commercially available monoclonal and polyclonal anti-THRα antibodies indistinguishably target both isoforms. This may have led to ambiguous biological interpretations in the past due to the opposing biological effects of the isoforms.

In this study, we present the first molecular tool to examine the expression levels of both THRα isoforms separately in vitro by fluorescent microscopy. The *pCMV-THRA-RFP-EGFP* splicing detector can be transfected into cultured cells using standardized transfection protocols. Fluorescence intensity allows relative quantification of the two isoforms. Longitudinal changes in isoform expression can be studied by time-lapse microscopy.

We have designed our *pCMV-THRA-RFP-EGFP* splicing detector in a way not to touch the intron 9 sequences (e.g., the 3′ UTR of exon 9b, all intronic sequences including the splice acceptor of exon 10) that Munroe and colleagues have shown to be critical for splicing regulation [10,11]. Once inside the target cell, pre-mRNA copies are transcribed from the plasmid and spliced by the cell’s intrinsic splicing machinery. Thus, factors (e.g., the composition of the spliceosome complex) that influence differential splicing and thus cell physiology can be studied and read out. We have confirmed that the intrinsic ratios of the two splicing isoforms are truly reported by the *pCMV-THRA-RFP-EGFP* splicing detector using a variety of methods in two different cell lines.

Expressing two different fluorescent proteins from the same plasmid to detect two splicing isoforms of the same protein has several advantages over the use of monochromatic reporter constructs. First, it allows the quantification of the full output of the target gene, which is particularly valuable for assessing alternative splicing events in single cells. Second, a single bichromatic reporter eliminates the variability that would be introduced by co-expressing two separate reporters in single cells. Finally, the mutual exclusivity of either RFP or EGFP expression from a pre-mRNA strand ensures that a change in the splicing pattern results in the gain of one fluorescent signal while the complementary signal is lost. This amplifies the relative signal change, as would be seen for the endogenous *THRA*.

Given the tight tissue- and time-dependent regulation of *THRA* splicing, it can be speculated that genetic alterations in the non-coding sequences of the *THRA* locus may disrupt the balance of THRα isoforms. This could potentially cause as-yet-undiagnosed rare inherited disorders in the realm of thyroid hormone signaling. This reporter can be used for high-throughput screening of libraries of cDNAs, shRNAs, and small molecules that can alter *THRA* splicing patterns and thereby potentially identify compounds capable of restoring correct splicing patterns in disease states. In addition, the general idea of fluorescently labeling the 3′ ends of multiple mRNA transcripts can be applied to a large number of genes, including some whose aberrant splicing leads to pathological conditions in humans [15,16]. The number of fluorophores to be inserted would be limited only by the spectral overlap(s) of their emission spectra. The main limitation of our *pCMV-THRA-RFP-EGFP* splicing detector lies in its relatively large size of 11.3 kbp. Larger plasmids are notoriously difficult to transfect into cells, would reduce cell survival after transfection, and have a higher probability of being ejected from the cells. However, appropriate selection and refinement of the transfection methods based on the specific cell line (e.g., lipofection, electroporation, viral gene transfer) may drastically improve uptake, expression, and survival rate. DNA delivery and improving transfection efficiency are currently major areas of research in gene therapy, and it is expected that more efficient methods for transfecting large plasmids will be readily available soon. Alternatively, the removal of exons and introns 7 and 8 could be considered to reduce the size of the *pCMV-THRA-RFP-EGFP* splicing detector. This should, however, be properly tested to confirm that the splicing ratios are kept constant and that no splicing regulatory sequences are lost in the intronic region.

In contrast to tagging both the full-length endogenous THRα isoforms, our *pCMV-THRA-RFP-EGFP* splicing detector was specifically designed as a plasmid to express truncated THRα1-RFP and truncated THRα2-EGFP as proxies for isoform expression without compromising or interfering with the functionality of the full-length endogenous THRα1 and THRα2 isoforms. In addition, the transient nature of the plasmid allows future applications for in utero electroporation and functional studies of THRα1 and THRα2 in vivo. The main drawback of this transient expression is that the non-integrating *pCMV-THRA-RFP-EGFP* splicing detector may be diluted during subsequent cell divisions, especially in fast-growing tissues and during development. Some possible solutions to this problem include transferring the modified *THRA* insert of the plasmid into the genome of a specific cell line (e.g., into a safe harbor locus), choosing cell lines that grow slower, and designing experiments with a reduced time window.

In conclusion, we have developed the first molecular tool that allows live-cell visualization and quantification of THRα1 and THRα2 isoforms in single cells by fluorescence microscopy. This tool will hopefully help the endocrinology community to address unanswered biological questions surrounding the complex function of thyroid hormone receptors (e.g., fully characterizing the role of THRα2) and then hopefully lead to the discovery of previously undiagnosed syndromes associated with non-coding *THRA* mutations.

## 4. Methods

### 4.1. Cloning of Truncated THRA Gene into Mammalian Expression Vector

The region of interest (7162 bp), spanning *THRA* exon 7 (GRCh38 chr17:40,086,707) to the 3′UTR region of *THRA* exon 10 (GRCh38 chr17:40,093,867), was amplified from healthy human donor DNA by long-range PCR using the Expand Long Template PCR System (Roche, #11681834011). We used a forward primer containing the Consensus Kozak sequence and a *Bgl*II restriction site upstream of the homology region of exon 7, and a reverse primer with a *Not*I restriction site downstream of the homology region of the 3′UTR region of exon 10. This region of interest (from now on called minigene) was first introduced into the pGEM^®^-T Easy amplification vector (Promega, #A1360) via TA cloning following the manufacturer’s protocol. The obtained plasmid was transformed into high-efficiency competent *Escherichia coli* NEB^®^ 5-alpha F’I^q^ (New England Biolabs, #C2992H) following the manufacturer’s protocol. Positive clones were validated by colony PCR with primers specific for the two ligation sites and by automated Sanger sequencing (BigDye^®^ Terminator v1.1 protocol, ABI 3500).

Next, the minigene was cloned into a mammalian expression vector containing the Cytomegalovirus (CMV) promoter (pEGFP-N3 vector, NovoPro, #V012019) via double-digestion cloning. Briefly, both expression vector and minigene (2 µg) were digested with *Bgl*II and *Not*I (5 units each) for 2 h at 37 °C and heat-inactivated for 10 min at 65 °C. Both digests were run on an agarose gel (Roche, #11388991001) and correct-sized bands (7 kb for the minigene and 4 kb for the linearized backbone of the expression vector) were cut out and purified using the NucleoSpin Gel and PCR Clean-up XS extraction kit by Macherey-Nagel (#740611.50). The purified minigene and the linearized backbone were then ligated at 16 °C for 16 h with a ratio of 3:1, respectively. As before, the resulting plasmid was transformed into high-efficiency competent *Escherichia coli* NEB^®^ 5-alpha F’I^q^ and positive clones were validated by colony PCR and by automated Sanger sequencing.

### 4.2. Cloning of EGFP and RFP into Expression Vector Containing THRA

Finally, we sequentially cloned EGFP into the place of the stop codon of exon 10 (GRCh38 chr17: 40,089,453-40,089,456) and RFP into the place of the stop codon of exon 9b (GRCh38 chr17: 40,093,379-40,093,382) using a modified protocol of the 2-PCR-step Megaprimer site-directed mutagenesis [13]. Briefly, in the first PCR step we amplified EGFP from pEGFP-N3 vector (NovoPro, #V012019) by long-range PCR using the Expand Long Template PCR System. We used EGFP-specific forward and reverse primers that were flanked by 100 bp homologous to the above-mentioned insertion site of the minigene on exon 10 (both upstream and downstream, see Table 1). For the second PCR step, we used the product of the first PCR step as a Megaprimer and the expression vector plasmid containing the minigene as a template. After amplification, the parental plasmid was removed by digesting with *Dpn*I (20 units) at 37 °C for 2 h. As before, the resulting plasmid was transformed into *Escherichia coli* NEB^®^ 5-alpha F’I^q^ and positive clones were validated by colony PCR and by automated Sanger sequencing. All these steps were then repeated for insertion of RFP that had been amplified from the pTagRFP-mito vector (Evrogen, #FP147) and 100 bp homology arms specific to exon 9b. The template of the second PCR step was the expression vector containing the minigene and EGFP on exon 10. The resulting plasmid was transformed into Escherichia coli NEB^®^ 5-alpha F’I^q^ and positive clones were validated by colony PCR and by automated Sanger sequencing.

### 4.3. Cell Culture: COS1

Fibroblast-like COS1 cells were purchased from ATCC (CRL-1650). Cells were cultured at 37 °C with 5% CO_2_ in Dulbecco’s modified Eagle’s minimal essential medium (DMEM; Gibco, #41966-029) supplemented with 15% heat-inactivated fetal bovine serum (FBS; Gibco, #10500-064) and 1% Pen-Strep (Gibco, #15140-122). A total of 20,000 COS1 cells in single cell suspension were seeded into 24-well plates (Corning, #3513). Then, 24 h after seeding, 1 µM of *THRA2*-specific FANA antisense oligonucleotides (AUM*block*^TM^ASO; AUM BioTech, LCC, Philadelphia, PA, USA; CTT CTC TCT CCT TTA CGA GAC), scrambled ASO, or fluorescently tagged scrambled ASO was added to the medium. Then, 24 h after ASO treatment, cells were transfected with 5 µg of *pCMV-THRA-RFP-EGFP* splicing detector using Lipofectamine 3000 (Invitrogen, #L3000001) diluted in Opti-MEM Reduced-Serum Medium (Gibco, #31985-062) as per the manufacturer’s guidelines and a 2:3 ratio of Lipofectamine/DNA. Unless stated otherwise, cells were incubated for 48 h before live imaging. For the time-lapse imaging experiments, cells were imaged every hour over 48 h, starting 24 h after initial transfection.

### 4.4. Cell Culture: N2A

Neuro-2a (N2A), mouse neuroblastoma cells were obtained from DSMZ, the German Collection of Microorganisms and Cell Cultures GmbH (#ACC 148). Cells were grown in 50% Dulbecco’s modified Eagle’s medium (DMEM; Gibco, #41966-029) and 50% Neurobasal medium (Gibco) containing 10% fetal calf serum (FCS; Gibco) and 1% penicillin and streptomycin (Gibco, #15140-122), and grown in 5% CO_2_ at 37 °C. Cells were transfected 24 h after plating with 5 µg of the *pCMV-THRA-RFP-EGFP* splicing detector using Lipofectamine 2000 (Invitrogen, #11668030), and a 2:3 ratio of Lipofectamine/DNA. Unless stated otherwise, cells were incubated for 48 h before live imaging. For the time-lapse imaging experiment, cells were imaged every hour over 17 h, starting 24 h after initial transfection.

### 4.5. Fluorescence Microscopy and QuPath Quantification

After either 48 or 24 h of incubation, cells were washed twice in phosphate-buffered saline (PBS; Gibco, #10010031) and 1 mL of FluoroBrite DMEM (Gibco, #A1896701) was added to the wells. Live cell fluorescence was recorded using a THUNDER Imager DMi8 with a Leica DFC9000 GT camera and the LAS(X) software (v 5.2.2, Leica CRL-1650). For time-lapse recording, cells were kept at 5% CO_2_ at 37 °C for the entire duration of the image acquisition. The imaging parameters (illumination, light intensity, aperture, exposure time, and camera sensitivity) were kept strictly constant for all channels and all recordings. For all the experiments and all conditions, we generated 5 visual fields (POVs) of 3 biological replicates with a 20× microscope lens on the green channel (THRα2), the red channel (THRα1), the far-red channel (CY5), and phase contrast. Microscopy images were analyzed using QuPath v0.5.1 software [14]. The signal from the green channel was used to quantify the mean fluorescence of *THRA2* splicing events. The signal from the red channel was used to quantify the mean fluorescence of the *THRA1* splicing events. Each mean signal was corrected by subtracting the background signal of the respective channel calculated on the same POV.

### 4.6. RNA Extraction and RT-qPCR

After live imaging, cells were washed with PBS, trypsinized (Trypsin-EDTA; Gibco, #25200056) and centrifuged. Pellets were snap frozen in liquid nitrogen and then stored at −80 °C. Total RNA was extracted from pellets using the NucleoSpin RNA XS kit by Macherey-Nagel (#740902.50). cDNA was prepared starting from 1 µg of total RNA using the SuperScript IV Reverse Transcriptase kit with oligo dT primers (#18090010). An amount of 50 ng of cDNA was used as template for each qPCR experiment (run in three independent technical replicates). qPCR experiments were run on a qTOWER^3^ G touch system (Analytik Jena, Jena, Germany) and proprietary software was used to generate raw C_t_ values. Raw data were processed in R. Standard curves for each primer pair (Table 2) were generated from a series of 1:10 dilutions of 10 steps from the respective PCR products and run under the same conditions. Efficiencies were calculated using the n = 7 dilutions that provided the best linear regression line. RNA levels of specific targets (*THRA1*, *THRA2*, RFP, and EGFP) were normalized against 18S C_t_ values using the ΔΔC_t_ method by Pfaffl et al. 2001 [17].

### 4.7. Data Analysis and Statistics

All raw data (both microscopy and qPCR) were processed and analyzed in R and Excel. Statistical comparisons were performed in GraphPad Prism v9.5.1 by tests specified in the legends of the respective figures. Graphs were generated either in R v4.3.0 (https://www.r-project.org/, accessed at 20 October 2024) or in GraphPad Prism v9.5.1. Unless stated otherwise, data were presented as mean ± standard deviation and a value of *p* < 0.05 was considered as statistically significant. For microscopy experiments, unless stated otherwise, we quantified n = 5 POVs for n = 3 biological replicates per condition (15 POVs total per condition). For qPCR experiments, each of the three biological replicates was quantified three times (n = 9 total Ct values per condition).

## Figures and Tables

**Figure 1 ijms-25-13512-f001:**
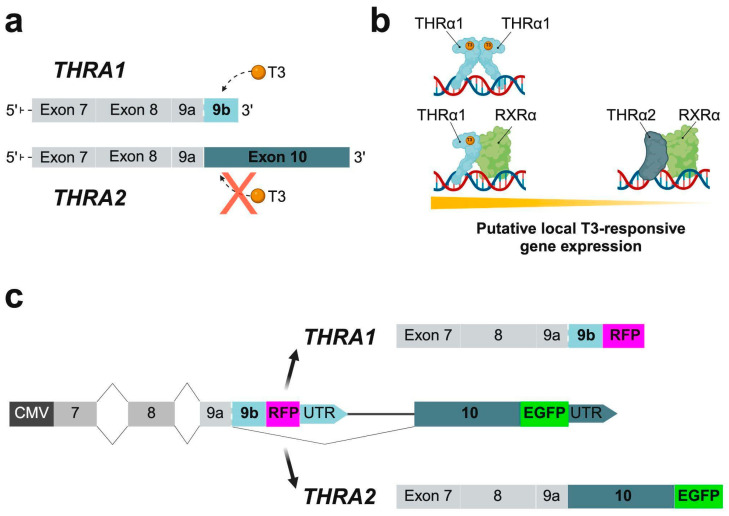
*THRA* isoforms and mechanism of action of the *pCMV-THRA-RFP-EGFP* splicing detector. (**a**) Schematic representation of the 3′ ends of the *THRA1* splicing isoform mRNA encoding THR*α*1 and of the *THRA2* splicing isoform mRNA encoding THR*α*2. The orange spheres represent the T3 ligand, the gray solid rectangles represent the exons common to both isoforms, and the light blue and dark blue solid rectangles represent the isoform-specific exons. The schematic shows that T3 can bind to THR*α*1 but not to THR*α*2. Adapted from [5]: (**b**) Schematic representation of local T3-responsive gene expression based on the relative abundance of THR*α*1 versus THR*α*2. Given the same amount of local T3, cell types that express more THR*α*1 will have higher T3-responsive gene expression levels compared to cell types that express more THR*α*2. Reproduced with permission from [5]: (**c**) Schematic representation of the *THRA*-specific gene sequence of the *pCMV-THRA-RFP-EGFP* splicing detector. Endogenous splicing mechanisms will produce either an N-terminally truncated THRα1 labeled with RFP or a truncated THRα2 labeled with EGFP at their C-termini.

**Figure 2 ijms-25-13512-f002:**
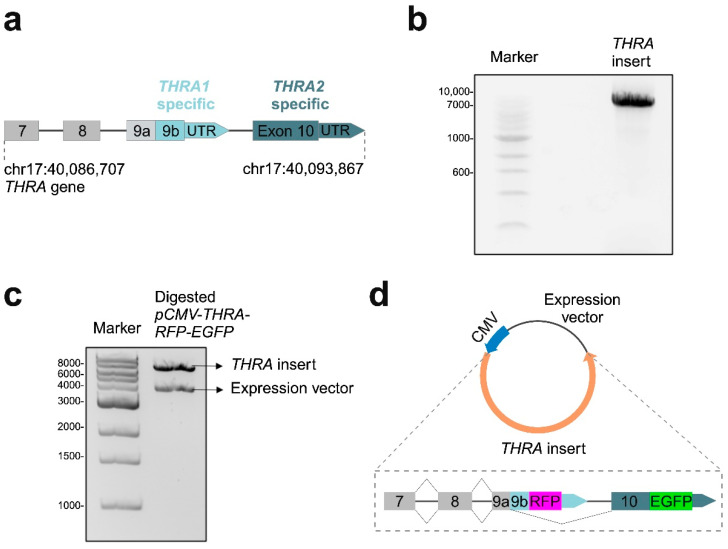
Cloning steps for the generation of the *pCMV-THRA-RFP-EGFP* splicing detector. (**a**) Schematic representation of the *THRA* region between exon 7 and the 3′ UTR region of exon 10 that we chose to integrate into our *pCMV-THRA-RFP-EGFP* splicing detector. *THRA1*-specific regions are shown in light blue, while *THRA2*-specific regions are shown in dark blue. (**b**) Agarose gel electrophoresis showing the correct 7300 bp PCR amplification product of *THRA* with flanking *Bgl*II and *Not*I sites. (**c**) Agarose gel electrophoresis showing the 7300 bp *THRA* insert and the 4000 bp linearized expression vector. (**d**) Schematic representation of the final *pCMV-THRA-RFP-EGFP* splicing detector with a CMV promoter to drive gene expression.

**Figure 3 ijms-25-13512-f003:**
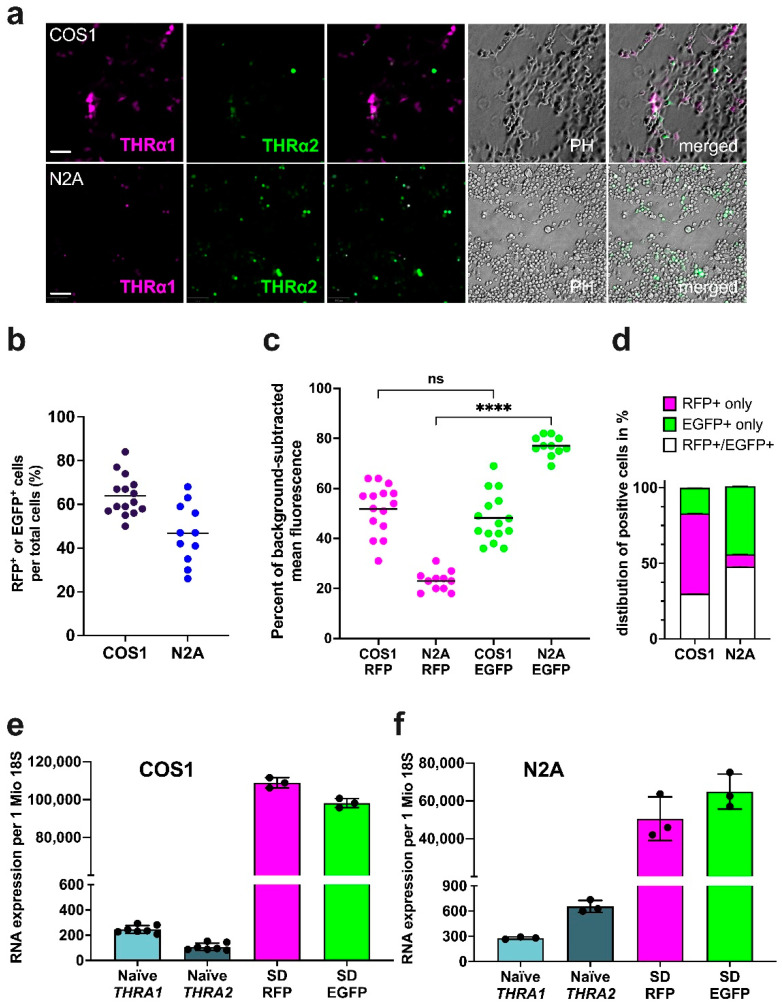
Validation of the *pCMV-THRA-RFP-EGFP* splicing detector in COS1 and N2A neuroblastoma cells. (**a**) Phase contrast and fluorescence images of COS1 and N2A neuroblastoma cells 48 h after transfection with the *pCMV-THRA-RFP-EGFP* splicing detector (SD). The RFP signal is a proxy for THRα1 and is shown in magenta. The EGFP signal is a proxy for THRα2 and is shown in green. Colocalization is shown in white. For each condition, experiments were performed in triplicate and 5 fields of view (POVs) were analyzed. One representative image per condition is shown. Scale bar: 100 µm. (**b**) Efficiency of *pCMV-THRA-RFP-EGFP* splicing detector transfection measured by EGFP^+^ and/or RFP^+^ cells in COS1 (black dots) and N2A neuroblastoma (blue dots) cells. Results of five POVs from three independent experiments per condition (n = 15 POVs per condition), presented as mean plus single dots. (**c**) Quantification of RFP (THRα1 proxy) and EGFP (THRα2 proxy) signals in COS1 and N2A neuroblastoma cells, shown as the percentage of mean fluorescence intensity after background subtraction. We observed a strong predominance of EGFP signal in N2A neuroblastoma cells and a slightly higher percentage of RFP compared to EGFP signal in COS1 cells. Results of five POVs from three independent experiments per condition (n ≥ 12 POVs per condition), presented as mean plus single dots. Kruskal–Wallis test and Dunn’s correction for multiple comparisons. ns = not significant; **** = *p* < 0.001. (**d**) Percentage composition of RFP^+^ and/or EGFP^+^ cells. RFP^+^-only cells (EGFP not detected) are shown in magenta, EGFP^+^-only cells (RFP not detected) are shown in green, and RFP^+^ and EGFP^+^ cells (both colors detected at various ratios) are shown in white. The bar graph shows a strong predominance of RFP^+^ in COS1 and a strong predominance of EGFP^+^ in N2A neuroblastoma cells. Results of five POVs from three independent experiments per condition (n = 15 POVs per condition), shown as mean ± standard deviation. (**e**) Normalized RNA levels of endogenous *THRA1* (light blue) and endogenous *THRA2* (dark blue) in naïve COS1 cells and RFP (proxy for THRα1, in magenta) and EGFP (proxy for THRα2, in green) in COS1 cells transfected with the *pCMV-THRA-RFP-EGFP* splicing detector (SD). As expected, CMV-promoted RFP and EGFP RNA levels were much higher compared to endogenously promoted *THRA1* and *THRA2* and the endogenous relative abundance of *THRA1* and *THRA2* was mirrored by the relative RFP-to-EGFP abundance. n ≥ 3. Results are expressed as mean ± standard deviation. (**f**) Normalized RNA levels of endogenous *THRA1* (light blue) and endogenous *THRA2* (dark blue) in naïve N2A neuroblastoma cells and RFP (in magenta) and EGFP (in green) in N2A cells transfected with the *pCMV-THRA-RFP-EGFP* splicing detector (SD). CMV-promoted RFP and EGFP RNA levels were orders of magnitude higher than endogenously promoted *THRA1* and *THRA2*; however, the relative *THRA1*-to-*THRA2* abundance in naïve samples was reflected by the relative RFP-to-EGFP abundance in transfected samples. n = 3. Results are expressed as mean ± standard deviation.

**Figure 4 ijms-25-13512-f004:**
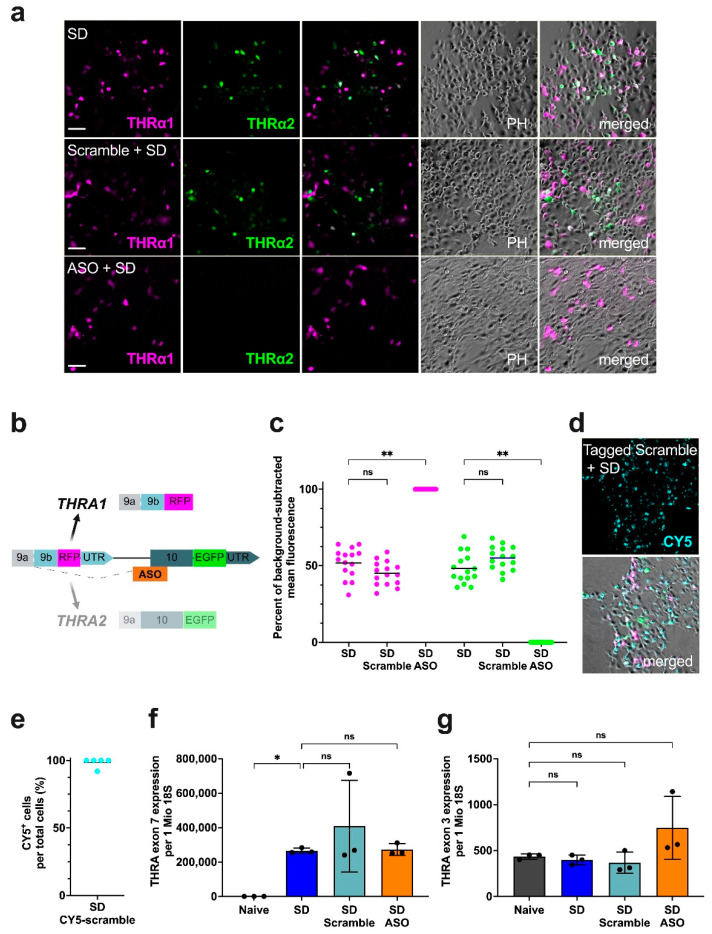
The *pCMV-THRA-RFP-EGFP* splicing detector can detect splicing perturbations in COS1 cells. (**a**) Phase contrast and fluorescence images of COS1 cells 48 h after transfection with the *pCMV-THRA-RFP-EGFP* splicing detector (SD). Cells were treated with either scrambled control ASO (scramble + SD) or *THRA2*-specific ASO (ASO + SD) 24 h prior to transfection. RFP signal is a proxy for THRα1 and is shown in magenta. EGFP signal is a proxy for THRα2 and is shown in green. Colocalization is shown in white. For each condition, experiments were performed in triplicate and 5 fields of view (POVs) were collected. One representative image per condition is shown. Scale bar: 100 µm. (**b**) Schematic representation of the mechanism of action of the *THRA2*-specific antisense oligonucleotides (ASOs). The ASO (in orange) is designed to bind to the region between *THRA* intron 9 and exon 10, sterically preventing *THRA2* splicing. (**c**) Quantification of RFP (THRα1 proxy) and EGFP (THRα2 proxy) signals, shown as a percentage of the mean fluorescence intensity after background subtraction. ASO treatment effectively blocked *THRA2* splicing, resulting in a loss of EGFP signal in ASO + SD samples and a statistically significant increase in RFP compared to the *pCMV-THRA-RFP-EGFP* splicing detector-only control. Scramble ASO showed similar levels of RFP and EGFP fluorescence compared to the *CMV-THRA-RFP-EGFP* splicing detector-only control. Results obtained from five POVs of three independent experiments per condition (n = 15 from five POVs randomly selected OVs per condition), presented as mean plus single data points. Kruskal–Wallis test and Dunn’s correction for multiple comparisons. ns, not significant; ** = *p* < 0.1. (**d**) Phase contrast and fluorescence images of COS1 cells pretreated with fluorescently labeled scrambled control ASOs. Images show the CY5 signal (cyan) as a proxy for tagged scramble ASO incorporation. (**e**) Efficiency of ASO uptake measured by the percentage of CY5^+^ cells. Results obtained from five POVs randomly selected from three independent experiments (n = 5 POVs), shown as mean plus single data points. (**f**) Normalized RNA levels of *THRA* exon 7, used as proxy of *pCMV-THRA-RFP-EGFP* splicing detector expression. No statistically significant difference was observed between the *pCMV-THRA-RFP-EGFP* splicing detector-only control (SD) and ASO-treated samples (scramble + SD; ASO + SD). All *pCMV-THRA-RFP-EGFP* splicing detector-transfected samples (SD; scramble + SD; ASO + SD) showed higher levels of exon 7 compared to naïve. n = 3 biological replicates (n = 3 C_t_ values for each biological replicate). Ordinary one-way ANOVA and Fisher’s LSD tests. ns, not significant; * = *p* < 0.05. (**g**) Normalized RNA levels of *THRA* exon 3, used as proxy for total endogenous *THRA* RNA expression. No statistically significant difference was observed between naïve and all transfected and treated samples (SD; scramble + SD; ASO + SD). n = 3 biological replicates (n = 3 C_t_ values for each biological replicate). Ordinary one-way ANOVA and Fisher’s LSD tests. ns, not significant.

**Figure 5 ijms-25-13512-f005:**
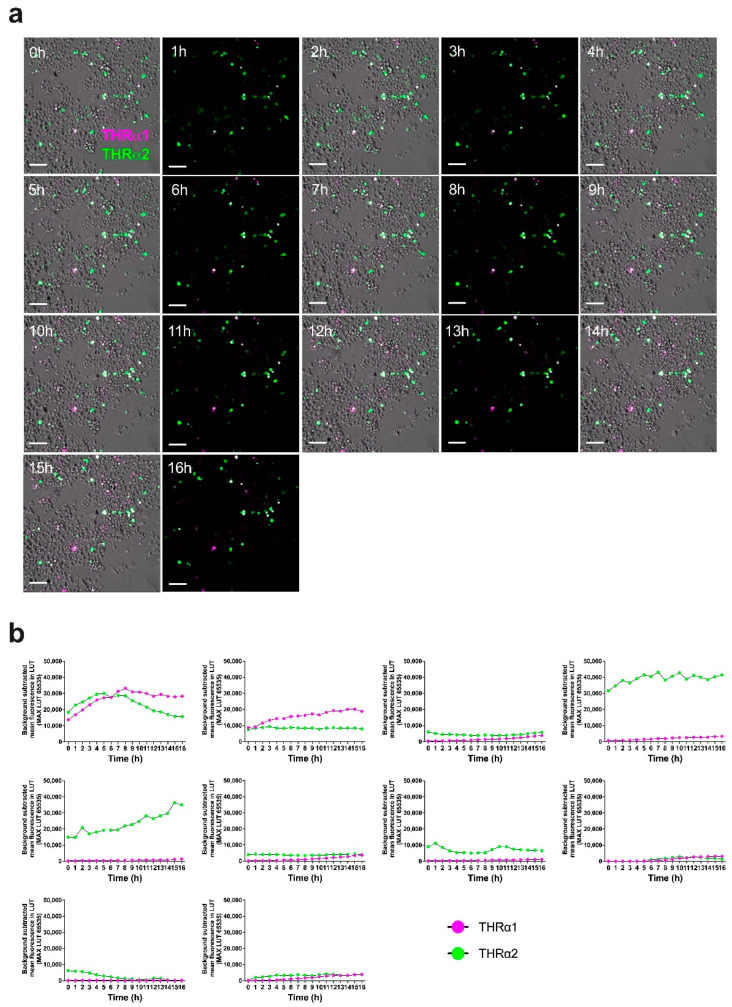
*pCMV-THRA-RFP-EGFP* splicing detector allows in vitro live time-lapse visualization and quantification of THRα1 and THRα2 isoforms in N2A neuroblastoma cells. (**a**) Phase contrast and fluorescence merged images of N2A neuroblastoma cells 24 h after transfection with the *pCMV-THRA-RFP-EGFP* splicing detector. The RFP signal is a proxy for THRα1 and is shown in magenta. The EGFP signal is a proxy for THRα2 and is shown in green. Co-expression is shown in white. Images were collected every hour for 16 h. Scale bar: 100 µm. (**b**) Single cell quantification of RFP (THRα1 proxy) and EGFP (THRα2 proxy) signals over time, shown as mean fluorescence intensity after background subtraction. The plots show the variability in THRα1 and THRα2 expression patterns in different single cells of the same cell line. n = 10 single cells.

**Figure 6 ijms-25-13512-f006:**
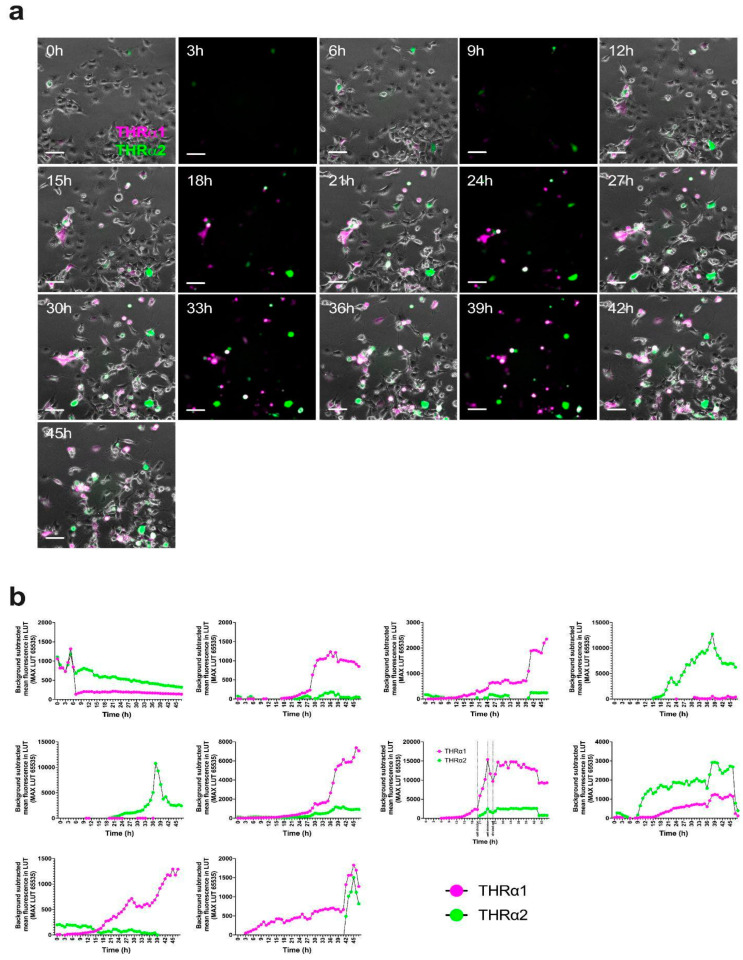
*pCMV-THRA-RFP-EGFP* splicing detector allows in vitro live time-lapse visualization and quantification of THRα1 and THRα2 isoforms in COS1 cells. (**a**) Phase contrast and fluorescence merged images of COS1 cells 24 h after transfection with the *pCMV-THRA-RFP-EGFP* splicing detector. The RFP signal is a proxy for THRα1 and is shown in magenta. The EGFP signal is a proxy for THRα2 and is shown in green. Co-expression is shown in white. Images were collected every hour for 48 h. Representative images every 3 h are shown. Scale bar: 100 µm. (**b**) Single cell quantification of RFP (THRα1 proxy) and EGFP (THRα2 proxy) signals over time, shown as mean fluorescence intensity after background subtraction. The plots show the variability in THRα1 and THRα2 expression patterns in different single cells of the same cell line. n = 10 single cells.

**Figure 7 ijms-25-13512-f007:**
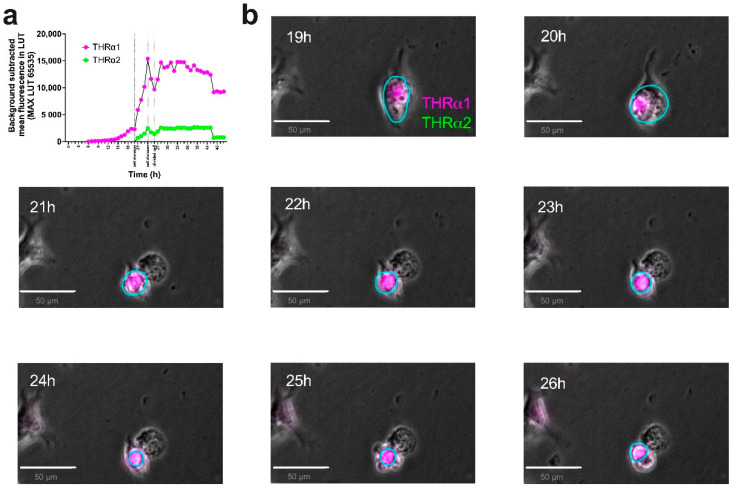
The *pCMV-THRA-RFP-EGFP* splicing detector allows in vitro live time-lapse visualization and quantification of THRα1 and THRα2 isoforms in COS1 cells during cell division. (**a**) Quantification of RFP (THRα1 proxy) and EGFP (THRα2 proxy) signals over time. The plot highlights an increase in THRα1 levels right before cell division and a subsequent drop immediately thereafter. n = 1 single cell (ROI shown in cyan). (**b**) Magnified view of Figure 6a phase contrast and fluorescence merged images of single COS1 cell undergoing cell division. The RFP signal is a proxy for THRα1 and is shown in magenta. The EGFP signal is a proxy for THRα2 and is shown in green. Co-expression is shown in white. Scale bar: 50 µm.

**Table 1 ijms-25-13512-t001:** Primers used for the Megaprimer site-directed mutagenesis.

Target	Orientation	Sequence (5′-3′)
*THRA* exon 10—EGFP	Forward	GGTCTGTGGGGAAGACGACAGCAGTGAGGCGGACTCCCCGAGCTCCTCTGAGGAGGAACCGGAGGTCTGCGAGGACCTGG
	Reverse	GCCCACGCTCCCAGCTTTCAGGCACCTCCTGCTCTTGGGGCAGAGGCCTGGGAGAAGGTATGGCACTCCTTCTCCTTCCC
*THRA* exon 9b—RFP	Forward	TCGGGGCCTGCCACGCCAGCCGCTTCCTCCACATGAAAGTCGAGTGCCCCACCGAACTCTTCCCCCCACTCTTCCTCGAG
	Reverse	AGAGAAGGGGTGTGGGGGGGTCTCCCTCAGCCCCCAGCTCTGCCCCTTCTCTCCAGGCTCCTCCCCACCAGCTCCGCACA

**Table 2 ijms-25-13512-t002:** Primers used for RT-qPCR experiments.

Target	Orientation	Sequence (5′-3′)
*18S* RNA	Forward	CATTCGAACGTCTGCCCTATC
	Reverse	CTCCCTCTCCGGAATCGAAC
*THRA* exon 3/4	Forward	CAGAGGAGAACAGTGCCAGGTC
	Reverse	CCTTGTCCCCACACACGAC
*THRA1* exon 9b	Forward	TGCTAATGTCAACAGACCGCT
	Reverse	CGATCATGCGGAGGTCAGTC
*THRA2* exon 10	Forward	TGCTAATGTCAACAGACCGCT
	Reverse	GCTGCCCCCTTGTACAGAAT
*THRA1*-RFP	Forward	TGCTAATGTCAACAGACCGCT
	Reverse	AATCAGCTCTTCGCCCTTAG
*THRA2*-EGFP	Forward	GAGGAACCGGAGGTCTGCG
	Reverse	CGTCGCCGTCCAGCTCGACCAG

## Data Availability

The *pCMV-THRA-RFP-EGFP* splicing detector has been deposited on AddGene repository for open use under catalog number #228537.

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
