# Peer review of "Bichromatic Splicing Detector Allows Quantification of THRA1 and THRA2 Splicing Isoforms in Single Cells by Fluorescent Live-Cell Imaging"

_ijms, 2024, doi:10.3390/ijms252413512_

Round 1
Reviewer 1 Report
Comments and Suggestions for Authors
This submission described a fluorescent detector for the quantification of THRA1 and THRA2 splicing isoforms in single cells. It was an interesting work and the experiments were meaningful. I recommend a major revision on the following points:
1. The most significant point was that, the detector in this work was an in vitro one. The in vivo application was difficult because there are original THRA1 and THRA2, while the isoforms could hardly replace the original ones. The authors should improve the significance of this work by trying to break this bottleneck. There should be at least some attempts and orientations.
2. In my opinion, since the tool was based on the modification of THRA1 and THRA2, the research target should be selected as the upstream regulators such as the behavior of T3. If possible, the authors might consider this point to extend their way thinking.
3. If the result included the WB experiment, the authors should provide the pictures of the original gels in the supporting materials.
4. The language use should be improved.
Comments on the Quality of English Language
The language use should be improved.
Author Response
Reviewer #1
This submission described a fluorescent detector for the quantification of THRA1 and THRA2 splicing isoforms in single cells. It was an interesting work and the experiments were meaningful. I recommend a major revision on the following points:
[1] The most significant point was that the detector in this work was an in vitro one. The in vivo application was difficult because there are original THRA1 and THRA2, while the isoforms could hardly replace the original ones. The authors should improve the significance of this work by trying to break this bottleneck. There should be at least some attempts and orientations.
Answer: We thank the reviewer for his/her time spent assessing our manuscript. In this study, we completely concentrate on the construction and evaluation of a tool to investigate differential splicing in living cells. The construct is not aimed at "replacing" the endogenous THRA transcripts. Actually, we do not want it to be biologically active since we did not want the reporter to interfere with cell metabolism. The reporter should only report on the different splicing events and should otherwise not interfere with transcription. The biological inactivity of the reporter allowed us to express it from a CMV promoter that guarantees high expression levels of the color-tagged exons in order to have a good microscopic signal.
[2] In my opinion, since the tool was based on the modification of THRA1 and THRA2, the research target should be selected as the upstream regulators such as the behavior of T3. If possible, the authors might consider this point to extend their way of thinking.
Answer: As mentioned above, this paper is not directed to elucidate intracellular targets of T3, be it up- or downstream. It is a mere technical paper describing a molecular tool to read out and quantify splicing events in living cells.
[3] If the result included the WB experiment, the authors should provide the pictures of the original gels in the supporting materials.
Answer: We did not perform a Western blot for this manuscript. The scans depicted in Figure 2b and 2c are from gel electrophoresis, not Western blot. The original raw scans of these electrophoretic gels have already been submitted alongside with our manuscript.
[4] The language use should be improved.
Answer: The English language has now been checked and corrected by a native English speaker

Reviewer 2 Report
Comments and Suggestions for Authors
The authors developed a novel bichromatic splicing detector plasmid for quantifying THRA1 and THRA2 isoforms in single cells using fluorescent live-cell imaging. The detector tool successfully visualized and quantified differential isoform expression in various cell types.
I think this manuscript could be accepted after minor corrections and I would leave some comments.
1. I just wondered, in this field, is it conventional to refer to novel fluorescent detectors with generic names, such as "Splicing Detector," rather than using specific initial names like FITC or DAPI? From the perspective of a researcher developing small molecule fluorescent probes, a name like "Splicing Detector" might feel somehow non-specific.
2. In Figure 3, is it proper to rigorously say that the magenta and green represent RFP+ 'only' and EGFP+ 'only', respectively? Maybe RFP+ 'dominant' and EGFP+ 'dominant' would be more clear. Also, the readers could be curious about what the ratio of magenta to green at the same position would appear as white colocalization.
Author Response
Reviewer #2:
The authors developed a novel bichromatic splicing detector plasmid for quantifying THRA1 and THRA2 isoforms in single cells using fluorescent live-cell imaging. The detector tool successfully visualized and quantified differential isoform expression in various cell types.
I think this manuscript could be accepted after minor corrections and I would leave some comments.
[1] I just wondered, in this field, is it conventional to refer to novel fluorescent detectors with generic names, such as "Splicing Detector," rather than using specific initial names like FITC or DAPI? From the perspective of a researcher developing small molecule fluorescent probes, a name like "Splicing Detector" might feel somehow non-specific.
Answer: We completely agree with the reviewer and have now renamed our splicing detector to "pCMV-THRA-RFP-EGFP". The same name has been used for the submission of the clone to AddGene (https://www.addgene.org/228537/)
[2] In Figure 3, is it proper to rigorously say that the magenta and green represent RFP+ 'only' and EGFP+ 'only', respectively? Maybe RFP+ 'dominant' and EGFP+ 'dominant' would be more clear. Also, the readers could be curious about what the ratio of magenta to green at the same position would appear as white colocalization.
Answer: The legend RFP+ only and EGFP+ only refers specifically to Figure 3d. Here we distinguished between cells in which both green and red (resulting white) signals could be detected and cells in which only either red (RFP) or green (EGFP) signals could be detected (after removing the background signal). We used the word “only” to differentiate between these two cases. The case where one cell had a dominant RFP signal and even the slightest EGFP signal was considered RFP+/EGFP+ and included in the white column.
We added the following text into the legend: "RFP+ only cells (EGFP not detected) are shown in magenta, EGFP+ only cells (RFP not detected) are shown in green, RFP+ and EGFP+ cells (both colors detected at various ratios) are shown in white."

Round 2
Reviewer 1 Report
Comments and Suggestions for Authors
The authors have improved the submission. I recommend the acceptance.